# Rare variant associations with birth weight identify genes involved in adipose tissue regulation, placental function and insulin-like growth factor signalling

Investigating the genetic factors influencing human birth weight may lead to biological insights into fetal growth and long-term health. We report analyses of rare variants that impact birth weight when carried by either fetus or mother, using whole exome sequencing data in up to 234,675 participants. Rare protein-truncating and deleterious missense variants are collapsed to perform gene burden tests. We identify 9 genes; 5 with fetal-only effects on birth weight, 1 with maternal-only effects, 3 with both, and observe directionally concordant associations in an independent sample. Four of the genes were previously implicated by GWAS of birth weight. *IGF1R* and *PAPPA2* (fetal and maternal-acting) have known roles in insulin-like growth factor bioavailability and signalling. *PPARG, INHBE* and *ACVR1C* (fetal-acting) are involved in adipose tissue regulation, and the latter two also show associations with favourable adiposity patterns in adults. We highlight the dual role of *PPARG* (fetal-acting) in adipocyte differentiation and placental angiogenesis. *NOS3* (fetal and maternal-acting), *NRK* (fetal), and *ADAMTS8* (maternal-acting) have been implicated in placental function and hypertension. To conclude, our analysis of rare coding variants identifies regulators of fetal adipose tissue and fetoplacental angiogenesis as determinants of birth weight, and further evidence for the role of insulin-like growth factors.

An improved understanding of the genetic contribution to birth weight may highlight mechanisms relevant to fetal growth restriction or overgrowth, as well as links between fetal growth and later disease[1]. Studies of rare, monogenic forms of diabetes have highlighted the key role of fetal insulin, since rare single gene mutations in the fetus that reduce insulin secretion are generally associated with a reduced birth weight. These include mutations that cause neonatal diabetes, where the severity of the insulin secretory defect correlates with the degree of reduction in birth weight[2]. In babies who produce no insulin at all due to pancreatic agenesis, or to complete loss-of-function (LoF)

mutations in the insulin gene, birth weights are 50% of those of a healthy term baby[3].

Family studies of mutations in the *GCK* gene have demonstrated the role of maternal genetic variants, which may influence birth weight independently of variants inherited by the fetus. Maternal heterozygous mutations in *GCK*, which reduce glucose sensing and cause fasting glucose to be regulated at a higher set-point, result in greater maternal glucose availability to the fetus, as glucose crosses the placenta. This causes the fetus, if it does not carry the mutation, to produce more fetal insulin and grow bigger in response[4]. Conversely, if the

✉ e-mail: R.Freathy@exeter.ac.uk

fetus carries a *GCK* mutation, its ability to sense the increase in glucose is impaired, causing reduced fetal insulin secretion and concomitant reduced fetal growth.

Genome-wide association studies of birth weight have identified > 200 regions of the genome where common variants (minor allele frequency (MAF) > 1%) are associated with birth weight[5,6]. Variants at the identified loci influence birth weight by direct effects of the fetal genotype, indirect effects of the maternal genotype acting on the intrauterine environment, or a combination of the two. Most of these variants likely influence growth via mechanisms that are independent of fetal insulin secretion[7]. Parent-of-origin effects in the fetus have been observed at several birth weight loci[6]. However, the causal genes at the vast majority of identified loci are unknown. Whole exome sequence (WES) data in biobank-scale studies offer the opportunity to identify genes that are causally related to birth weight, and the potential to uncover new mechanisms of importance for the regulation of fetal growth.

In this work, we performed exome-wide association studies (ExWAS) of rare variant (MAF < 0.1%) gene burden with birth weight in up to 234,675 UK Biobank participants who reported their own birth weight (fetal variants), and up to 181,883 female UK Biobank participants who reported the birth weight of their first child (maternal variants). The identified genes highlight key roles for adipose tissue regulation and fetoplacental angiogenesis, in addition to the role of insulin-like growth factor bioavailability and signalling, in regulating human birth weight.

## Results

We identified 8 genes in which rare (MAF < 0.1%) deleterious LoF fetal variants, defined as either high-confidence protein truncating variants, or missense variants with a CADD score ≥ 25, were associated with birth weight at exome-wide significance (BOLT-LMM $P < 1.64 \times 10^{-6}$): *ACVR1C, IGF1R, INHBE, NOS3, NRK, NYNRIN, PAPPA2, PPARG* (Fig. 1, Supplementary Fig. 1–3, Supplementary Data 1). In the analysis of maternal variants and offspring birth weight, 3 genes showed associations at exome-wide significance (BOLT-LMM $P < 1.58 \times 10^{-6}$). These included 2 genes also identified in the fetal analysis (*IGF1R* and *NOS3*) and

additionally *ADAMTS8* (Fig. 1, Supplementary Fig. 4–5, Supplementary Data 1). We also confirmed all of these significant associations by conducting independent analyses using REGENIE (see "Methods") and we did not observe any method-related biases (Supplementary Fig. 1, Supplementary Data 1).

Since maternal and fetal genotypes are correlated (r = 0.5), we explored whether the observed associations represented fetal effects, maternal effects, or both. To do this, we used a weighted linear model (WLM) to approximately condition the fetal effect on the maternal genotype, and vice versa. Five out of nine associations (*ACVR1C, INHBE, NRK, NYNRIN, PPARG*) showed evidence of only fetal-genotype effects. *IGF1R, PAPPA2,* and *NOS3* were classified as both fetal- and maternal-acting, with rare variants in all three genes associating with a lower birth weight in both cases. *ADAMTS8* was the only gene classified as only maternal-acting (Fig. 2a, Supplementary Data 2).

For fetal genes associated with birth weight, we explored whether they might act in a sex-specific manner. We found a nominally stronger effect for fetal *PPARG* variants on birth weight in females (beta$_{Female}$ = -1.710 [0.340] SDs) than in males (beta$_{Male}$ = -0.177 [0.592] SDs, $p_{het}$ = 2.46 × 10$^{-02}$), albeit only 3 male carriers were identified (Fig. 2b, Supplementary Data 3).

### Confirmation of exome associations

We aimed to replicate our associations in independent data on up to 45,622 Icelandic genomes. Despite the substantially smaller sample size than our discovery sample, we observed supportive evidence for the same type of rare LoF exome variants on birth weight for the majority of our associations (10/11 showed directional consistency; Sign test P = 0.01), with three genes associated at P < 0.05; *INHBE, NRK* and *PAPPA2* (Supplementary Fig. 6, Supplementary Data 4).

To identify whether any of the birth weight genes identified in our exome analyses are also supported by common variant associations, we examined the latest maternal and fetal GWAS summary statistics for birth weight[6] and looked for GWAS signals within 300 kb of our identified genes. We observed fetal GWAS signals proximal to *ACVR1C, IGF1R,* and *PAPPA2,* and there was also a maternal GWAS signal proximal to *NOS3* (Supplementary Fig. 7). In all cases the lead associated

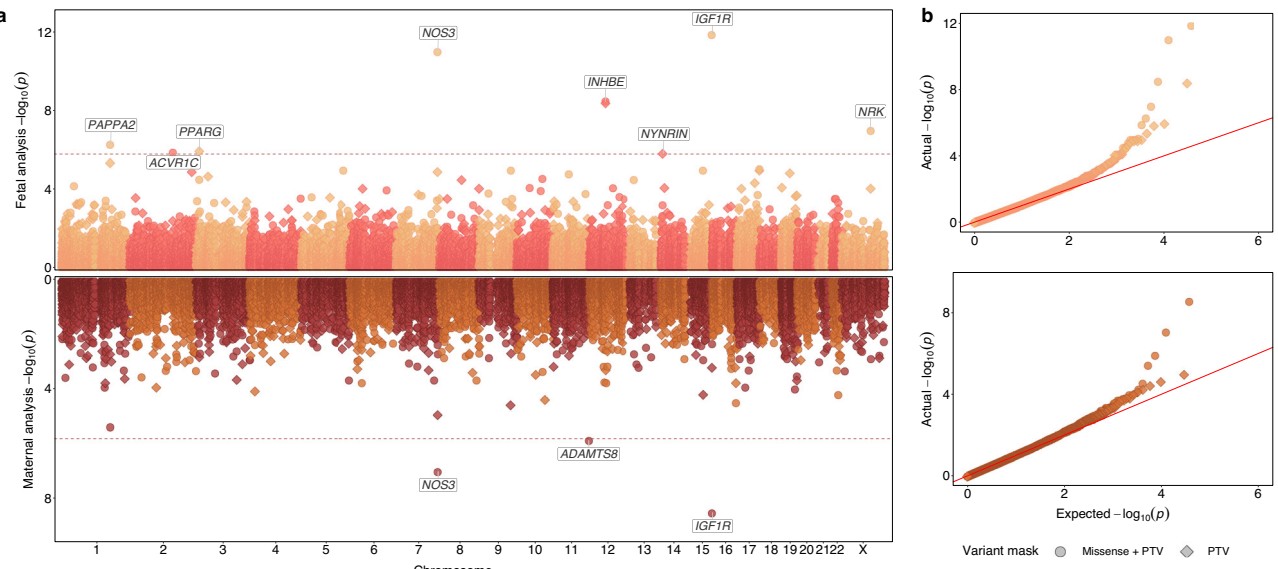

**Fig. 1 | Exome-wide rare fetal or maternal variant gene burden associations with birth weight. a** Miami plot showing gene burden test results from BOLT-LMM for birth weight, with the fetal exome-wide analysis (up to *n* = 234,675) on the top panel and the maternal exome-wide analysis (up to *n* = 181,883) on the bottom panel. Gene associations passing exome-wide significance, at the multiple-test corrected thresholds of $P < 1.64 \times 10^{-6}$ in the fetal analysis and $P < 1.58 \times 10^{-6}$ in the maternal analysis, are labelled (two-sided test). The two rare (MAF < 0.1%) variant collapsing masks are indicated by point shapes. **b** QQ plots for exome-wide gene burden associations from BOLT-LMM. Relevant data are included in Supplementary Data 1.

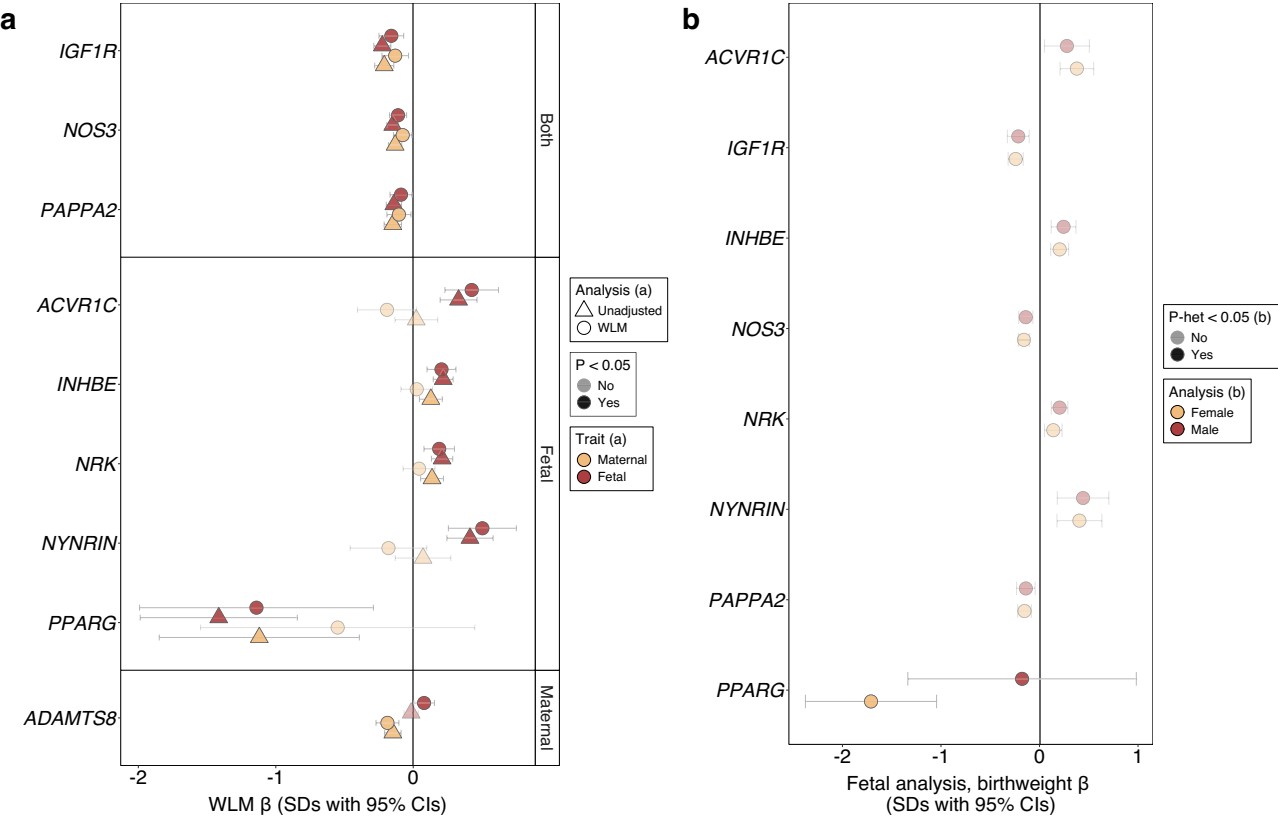

**Fig. 2 | Nine genes with rare variant associations with birth weight. a** Weighted-linear models (WLMs) approximately conditioned the fetal effect on the maternal genotype, and vice versa. Points indicate the mean effect estimates and accompanying 95% CIs, in SDs. Marker opacity indicates associations at $P < 0.05$ (two-sided). **b** Sexually-dimorphic effects of the associated genes on birthweight. Markers are coloured yellow or red to indicate female- or male-only models. Points indicate the mean effect estimates and accompanying 95% CIs, in SDs. Marker opacity indicates sexually dimorphic associations ($P < 0.05$, two-sided). Relevant data are included in Supplementary Data 2 and 3, accordingly.

variants were located within these genes (Supplementary Data 5) and proximity has been shown to be a good predictor of the causal gene at a GWAS locus[8,9]. The lead SNPs at the *IGF1R* and *NOS3* loci were also eQTLs for these genes, with directions of effect concordant with the exome associations, i.e., the alleles associated with lower expression (LoF-like) were also associated with lower birth weight[10]. Of these four GWAS signals, *IGF1R*, *ACVR1C*, and *PAPPA2* were classified as having fetal only effects by Juliusdottir et al.[6]. The SNP near *NOS3* was not classified by Juliusdottir et al.[6], but the *NOS3* lead SNP was classified by Warrington et al.[5] as having only maternal effects. We did not observe attenuation of the rare variant associations with birth weight when conditioning on the proximal GWAS sentinel SNPs (Supplementary Data 6). Hence, these common variant associations provide independent support for these genes in the regulation of birth weight.

To assess evidence for additional candidate genes influencing birth weight at GWAS loci, which did not pass our stringent exome-wide significance thresholds, we looked at all genes within 300 kb of a reported GWAS locus and applied a Benjamini-Hochberg correction for multiple testing. The results are shown in Supplementary Data 7. In the fetal variant analysis we identified a further eight candidate genes (*CDK6, HGS, PAPPA, PHF19, PLAG1, PLCE1, PTEN* and *SKP2*), and in the maternal variant analysis we identified one additional candidate gene (*LMNA*). At six of these genes (*CDK6, PAPPA, PHF19, PLAG1, PLCE1* and *SKP2*) a GWAS lead SNP was located within the same gene, while the lead SNP within *PLCE1* was also a missense variant, further adding to the evidence that non-synonymous coding variation in these genes affects birth weight.

Finally, we explored whether the biology implicated by the exome-associated genes was consistent with that from GWAS studies. To do

this, we performed a pathway enrichment analysis, using genes proximal to the GWAS signals from Juliusdottir et al.[6], some of which were also highlighted by our exome analyses as detailed above. We saw enrichment for 185 GO pathways in the fetal GWAS of birth weight and 41 pathways in the maternal GWAS, with 21 pathways being enriched in both GWAS (fetal and maternal, Supplementary Data 8). In line with biology implicated by our exome associations, we observed significant enrichment for genes in multiple pathways implicated in insulin and growth factor response across both fetal and maternal GWAS genes and circulatory system development for fetal GWAS genes, among others.

## Gene burden associations with related traits
To understand the wider phenotypic impacts of birth weight-associated rare genomic variation, we performed phenotypic association lookups in birthweight-related traits using data from the UK Biobank (see "Methods").

We observed associations for rare LoF variants in *IGF1R* and *PAPPA2*, both of which are known regulators of insulin-like growth factor (IGF) regulation and bioactivity[11–15], with adult circulating IGF-1 levels, adult height and childhood height and body size (Supplementary Data 9). The *INHBE, NRK, NOS3* and *NYNRIN* genes were also associated with adult height. In all of these genes, apart from *INHBE*, carrying a rare LoF variant was associated with directionally concordant effects on birth weight and adult height, i.e., higher birth weight and higher adult height or vice versa.

Furthermore, we observed associations between variants in *ADAMTS8* and *NOS3* and adult blood pressure (Supplementary Data 9). For both of these genes, rare LoF variants were associated with lower birth weight and higher blood pressure.

Finally, several of the birth weight associated genes were also associated with measures of adiposity in adulthood, including waist-hip ratio (WHR) adjusted for BMI, and body fat percentage for rare LoF variants in *ACVR1C*, *INHBE* and *PPARG* (accordingly, Supplementary Data 9). These are further explored below.

### *ACVR1C*, *INHBE* and favourable adiposity

*ACVR1C* (birth weight: beta$_{fetal}$ = 0.330 SDs, $p = 1.4 \times 10^{-6}$, N = 200 carriers, Fig. 3a) encodes a type I receptor for the TGF-beta family of signalling molecules. It is predominantly expressed in adipose tissue[10] and its primary function in metabolic regulation is to limit catabolic activities and preserve energy[16]. Low-frequency heterozygous mutations in humans have been linked to a favourable metabolic profile including lower WHR adjusted for BMI and protection against T2D[17,18]. *INHBE* (beta$_{fetal}$=0.218 SDs, $p = 3.5 \times 10^{-9}$, N = 687 carriers, Fig. 3b) encodes the inhibin subunit beta E of activin E, a growth factor belonging to the TGF-beta family. While this gene is predominantly expressed in the liver[10], it does not appear to be necessary for normal liver function[19], but rather acts as a negative regulator of energy storage in peripheral adipose tissue. Similarly to *ACVR1C*, a candidate receptor, heterozygous LoF carriers exhibit lower WHR adjusted for BMI and a favourable body fat distribution and metabolic profile[20,21]. However, recent mouse studies[22,23] showed that complete ablation of INHBE or ACVR1C leads to uncontrolled and pathological levels of lipolysis (see Discussion). We found that rare LoF variants in *ACVR1C* and *INHBE* are associated with increased birth weight and decreased WHR, indicative of a metabolically favourable adiposity pattern in adulthood (Supplementary Data 9).

Two genetics studies[20,21] showed that the association between rare LoF variants in *INHBE* and WHR is primarily due to a splice acceptor variant (12:57456093:G:C) that substantially reduces the expression of the gene, attributable to either a change in secretion and/or protein stability. They also reported that the favourable adiposity effect of this gene is almost entirely attenuated in the absence of this variant (gene-burden $p = 2.01 \times 10^{-8}$ with splice variant, $p = 0.34$ without[21], which we also observe for WHR in our data (Supplementary Data 10). However, while this variant is also strongly associated with higher birth weight ($p = 5.4 \times 10^{-6}$, Fig. 3b), our observed birth weight association with *INHBE* does not fully attenuate in the absence of this variant (gene-burden $p = 4.6 \times 10^{-9}$ with splice variant, $p = 6.3 \times 10^{-5}$ without, Supplementary Data 10).

### *PPARG* in birth weight, adipogenesis, and placental formation

*PPARG* (birth weight: beta$_{fetal}$ = −1.416 SDs, $p = 1.2 \times 10^{-6}$, N = 11 carriers, Fig. 4b) encodes the peroxisome proliferator-activated receptor PPAR-gamma, which acts as a master regulator of adipogenesis. Heterozygous LoF variants in humans have been linked to familial partial lipodystrophy (FPLD) characterised by loss of subcutaneous fat from the extremities and metabolic abnormalities such as insulin resistance and hypertriglyceridemia[24-27]. Individuals with FPLD due to inherited heterozygous variants have also been shown to have low birthweight for their gestational age, when in the absence of maternal diabetes, hypertension or hypertriglyceridemia (e.g., when the *PPARG* variants are paternally inherited)[28]. In addition to its high expression in adipocytes, *PPARG* is highly expressed in trophoblasts in both rodent and human placentas[29] where it appears to play an important role in placental formation and angiogenesis by regulating the expression of

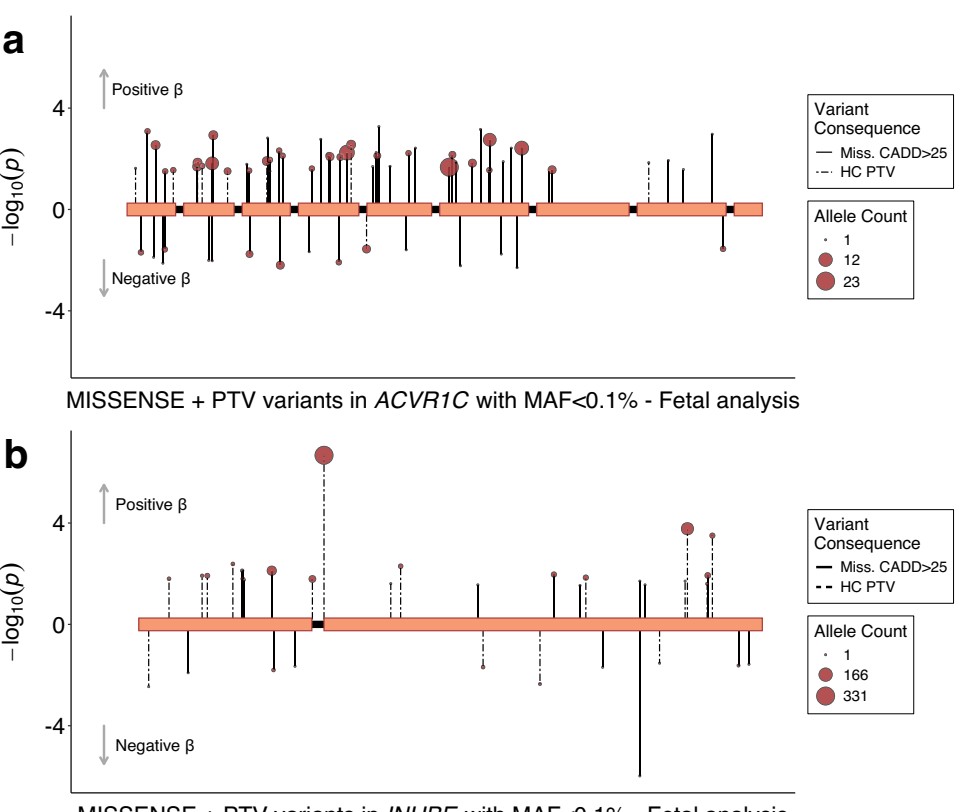

**Fig. 3 | Rare variant associations at *INHBE* and *ACVR1C* with fetal birth weight in the UK Biobank.** Fetal variant-level associations from BOLT-LMM between *ACVR1C* (**a**), *INHBE* (**b**) and birth weight. Included variants had a minor allele frequency (MAF) < 0.1% and were annotated to be damaging variants defined as high-confidence protein truncating variants (PTV) or missense variants with a CADD score ≥ 25. Each variant is presented as an individual line extending to its association *p*-value (-log$_{10}$) in the direction indicating the direction of effect on birth weight in variant carriers. Dashed lines indicate PTVs and solid lines indicate missense variants. The point size indicates the number of carriers of each variant (i.e., the allele count).

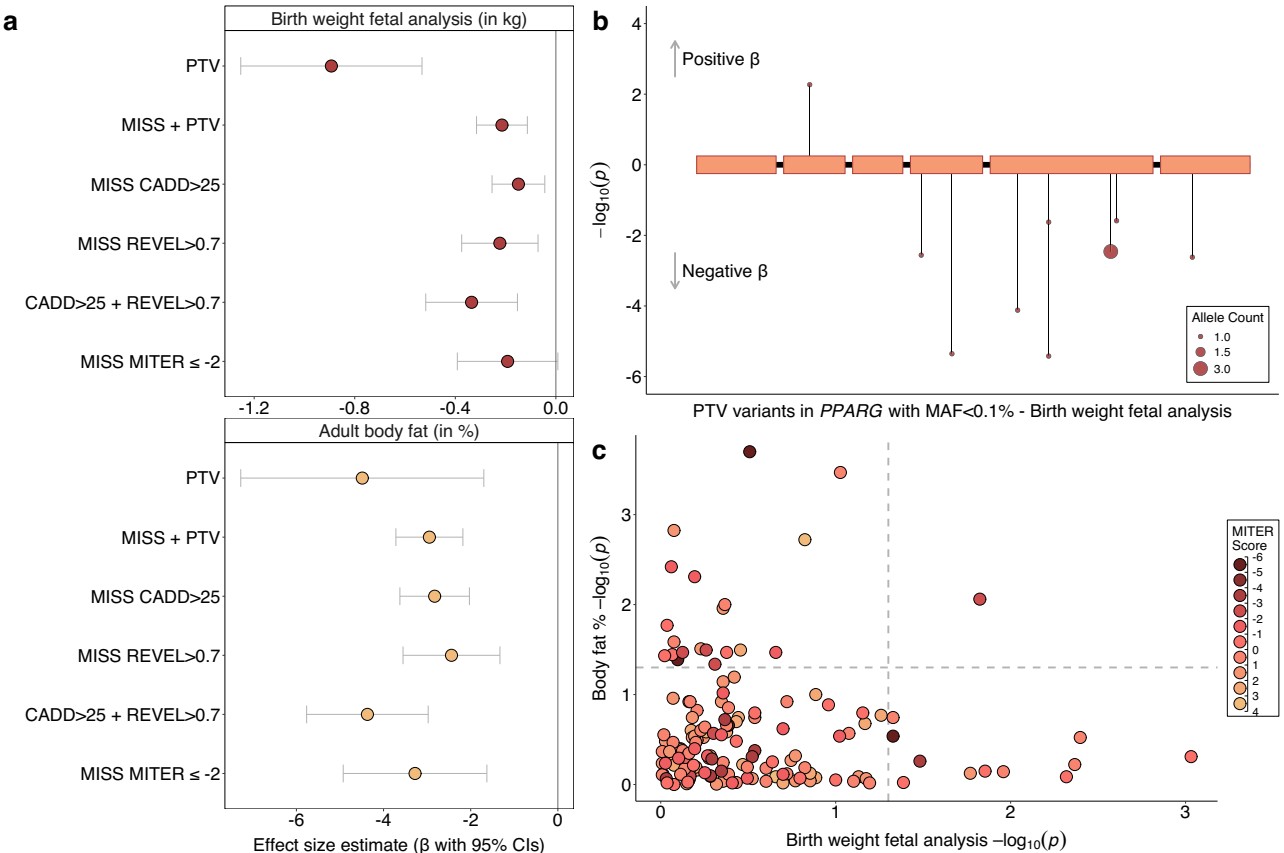

**Fig. 4 | Associations with rare LoF variants in *PPARG*. a** Different (fetal) variant collapsing masks and their associations with birth weight (top) and adult body fat percentage (bottom). Points indicate the mean effect estimates and accompanying 95% CIs. **b** The genomic location and associations from BOLT-LMM with birth weight for all qualifying fetal high confidence PTVs within *PPARG*, in the discovery analysis. **c** Scatterplot of association *P*-values from BOLT-LMM for birth weight (fetal variants) and body fat percentage, for all rare missense variants in PPARG annotated with a MITER score and found in the UK Biobank population. Marker colour indicates the variants MITER score, with scores ≤ −2 indicating lipodystrophy-level MITER scores. Relevant data is included in Supplementary Data 1.

proangiogenic factors such as proliferin (*Prl2c2*) and vascular endothelial growth factor (VEGF)[30]. Its depletion results in embryonic lethality in mice due to placental dysfunction[31,32]. In humans, PPARγ agonist activity was reduced in the blood and placentas of patients with severe preeclampsia, compared to those with normal pregnancy[33].

To examine if our observed birth weight association might correspond to known effects of PPARγ function in adipogenesis, we annotated rare missense variants in *PPARG* using Missense InTerpretation by Experimental Response (MITER) score data[34], as the gold-standard indicator of their lipodystrophy-causing potential. Although MITER scores are unavailable for PTV variants, for which we observed the strongest birth weight association, we note that 8 out of our included 9 PTV variants are classified as pathogenic based on ACGS[35]. *PPARG* missense variants with a lipodystrophy-causing MITER score (≤−2) showed a similar magnitude collective association with lower adult body fat percentage than our other prediction based missense variant masks ($p = 9.70 \times 10^{-5}$, Fig. 4a, Supplementary Data 11). We found only weak evidence of association between *PPARG* MITER score variants and birth weight ($p = 0.06$), although the effect size estimate was not significantly different from the other missense masks (Fig. 4a, Supplementary Data 11). There was also no discernible correlation between *PPARG* variant effect estimates on birth weight, body fat percentage and their corresponding MITER scores (Pearson correlations: $R_{BW-MITER} = 0.026$, $R_{BF\%-MITER} = 0.187$, $R_{BF\%-BW} = 0.101$, Fig. 4c). This data does not support an impact on birth weight mediated by

adipogenesis, but further data is needed to link rare *PPARG* variants to placental morphology and function.

As with *PPARG*, the *ADAMTS8, NRK, NYNRIN*, and *NOS3* genes have also been implicated in placental development and function. Specifically, *ADAMTS8* shows strong placental expression in the early stages of gestation and has been implicated in endometrium decidualization and trophoblast invasion[36–39]. *NRK* encodes a serine/threonine kinase of the germinal centre kinase family and is expressed in skeletal muscle during development as well as the placenta, specifically the spongiotrophoblast layer, a fetus-derived region of the placenta[40–42]. Nrk-deficient mouse fetuses display enlarged placentas, a likely result of enhanced trophoblast proliferation due to upregulation of AKT phosphorylation[43], as well as a higher birth weight and delayed delivery through yet unknown mechanisms[42]. Common variants in *NOS3* have been previously associated with the risk of developing preeclampsia[44], although no association was found at this locus in a large-scale preeclampsia GWAS[45]. Finally, *NYNRIN* has been suggested as a modulator of trophoblast invasion and linked to the evolutionary emergence of the placenta[46].

## Comparison of burden test results with birth weight effects of pathogenic *GCK* variants

We performed a sensitivity analysis to assess how bioinformatically-informed variant classification, as used in our ExWAS, compared with the clinical annotation of known variants. We focused on the *GCK* gene because it harbours pathogenic variants which are known to influence

fetal growth when present in either the mother's or the child's genome, and which we hypothesised would be common enough to be detectable with our sample size, given the published effect sizes and prevalence[4,47]. Mothers with pathogenic *GCK* variants have a stably raised fasting glucose level, which results in greater glucose availability to the fetus, and increased birth weight due to fetal insulin secretion, which is raised in response to higher maternal and consequently fetal glucose. Conversely, fetal growth is reduced in babies who carry pathogenic *GCK* variants due to their reduced insulin secretion. These birth weight effects are masked if both mother and fetus carry the same mutation[4]. We annotated and combined pathogenic variants in *GCK* using clinical guidelines[35] and observed evidence of both fetal and maternal genetic associations with birth weight in the expected directions (beta$_{fetal}$ = −0.311 SDs, $p$ = 0.002, N = 86 carriers; beta$_{maternal}$ = 0.430 SDs, $p$ = 0.001, $N$ = 53 carriers; Supplementary Fig. 8; Supplementary Data 12). In comparison, our gene burden associations with rare predicted LoF variants in *GCK* were more modest but in the expected directions (Supplementary Fig. 8; Supplementary Data 12). We found 63 of the known pathogenic *GCK* variants present in the UK Biobank WES dataset. Out of 16 variants in the PTV mask, 14 were annotated as pathogenic and of 55 variants in the Missense + PTV mask, 33 were annotated as pathogenic. The remaining pathogenic variants were annotated as non-deleterious and were not included in the masks.

## Discussion

In this study, we used whole exome sequencing data from the UK Biobank cohort to understand the influence of rare genetic variants on human birth weight. We aimed to identify genes harbouring rare variants that impact birth weight when carried by either the fetus or the mother. We identified a total of 8 genes with rare fetal LoF variant effects on birth weight, two of which, *IGF1R* and *NOS3*, showed some evidence of maternal effects. We identified one additional maternal effect gene, *ADAMTS8*. These findings were largely consistent in an independent cohort of Icelandic genomes. Four of the identified genes (*IGF1R*, *PAPPA2*, *ACVR1C*, and *NOS3*) were proximal to lead SNPs from the largest and most recent birth weight GWAS meta-analysis[6], providing further support that they are causal genes involved in birth weight regulation.

Among the genes associated with birth weight, several have well-established links with fetal growth. *IGF1R* and *PAPPA2* encode key components of IGF bioavailability and signalling. IGF1R is a transmembrane receptor tyrosine kinase activated by IGF1 (and also IGF2 and insulin), which mediates anabolic effects. Rare human homozygous, heterozygous or compound heterozygous mutations in *IGF1R* cause intrauterine growth restriction, reduced postnatal growth, short stature and microcephaly[48–51]. *PAPPA2* encodes a metalloproteinase that regulates the bioavailability of IGFs. Rare homozygous mutations in *PAPPA2* are reported to cause severe short stature but with unclear effects on birth weight (reduced in 4 of 6 affected children)[52], while the protein originates primarily from trophoblasts in the early placenta[53,54]. Our findings demonstrate that rare, damaging variants in *IGF1R* and *PAPPA2* contribute to population variation in birth weight, as well as in childhood and adult height.

The association between the rare, damaging fetal variants in *IGF1R* and greater type 2 diabetes risk provides a potential mechanistic insight into the well-documented link between this later-life disease and lower birth weight[1]. *IGF1R* is central to the development of several key tissues for glucose metabolism, including pancreatic islets, adipose tissue and skeletal muscle[55], and in addition, loss of *IGF1R* function may lead to compensatory increases in growth hormone levels, which are associated with insulin resistance[56]. In demonstrating that lower birth weight and type 2 diabetes are two phenotypes of the *IGF1R* LoF genotype, our finding is consistent with the principle of the fetal insulin hypothesis[57], while providing novel data to suggest that the

epidemiological association between the two phenotypes may involve fetal IGF1 as well as fetal insulin. The mechanism underlying the association we observed between maternal rare, damaging *IGF1R* variants and lower offspring birth weight after adjustment for fetal genotype is, however, less clear because we would expect such variants to increase birth weight if they result in higher maternal insulin resistance with concomitant higher glucose availability. However, there are some reported cases of heterozygous IGF1R mutations leading to hypoglycemia[58]. We recommend that this result is confirmed in well-powered samples of mother-child pairs before potential mechanisms are investigated.

*PPARG*, *INHBE*, and *ACVR1C* displayed fetal effects on birthweight and are involved in adipose tissue differentiation and regulation. Recent genetics studies implicated rare LoF variants in *INHBE* in adult favourable adiposity traits, i.e., lower WHR adjusted for BMI[20,21]. They also highlighted a splice variant (12:57456093:G:C) which affects either protein secretion or stability, as underpinning the favourable adiposity association with this gene. We found that the birth weight association with *INHBE* largely remains, even in the absence of this splice variant. This could indicate that some variants within *INHBE* may exert a role in early (*in utero*) adipogenesis, while others may do so throughout the life course, but further studies need to confirm this hypothesis.

Our identification of rare LoF variants in two genes, *ACVR1C* and *INHBE*, that have fetal associations with higher birth weight and also a more metabolically favourable body fat distribution in adulthood is in line with recent findings that common genetic variants in the fetus, which predispose to higher metabolically favourable adiposity in adulthood, are also associated with higher birth weight. Variants with greater effects on adiposity have greater effects on birth weight[59] and we therefore hypothesise that the birth weight effects of *ACVR1C* and *INHBE* are due primarily to fetal fat accretion rather than effects on lean mass. A fetal genetic predisposition to greater birth weight and later favourable body fat distribution may underlie epidemiological associations between greater skinfold thickness at birth and favourable metabolic outcomes in early childhood, especially when controlling for exposure to maternal glucose levels *in utero*[60]. Further studies should assess the contributions of rare variation in *ACVR1C* and *INHBE* to the fat and lean mass components of birth weight.

More generally, the effects of loss of INHBE on Alk7 (ACVR1C) signalling on metabolic health appear to be complex and dose-dependent. Humans haploinsufficient for either of these genes have a healthy metabolic profile and reduced risk of cardiometabolic disease accompanied by a favourable distribution of body fat. However, it has been shown that the total absence of either INHBE or ACVR1C from conception results in uncontrolled adipocyte lipolysis, ectopic fat distribution and insulin resistance, at least in mice[22,23]. This discrepancy may be explained by the fact that Alk7 signalling in adipose cells suppresses the expression of both PPARγ and key molecular elements of the lipolytic machinery. We suggest that at modest levels of INHBE/Alk7 deficiency, such as occurs when only one functional allele is present, the effect on PPARγ is dominant and carriers of such mutations develop more subcutaneous adipocytes as a result of the enhanced PPARG tone during development. Ultimately, this is metabolically beneficial as it will tend to protect the individual from any effects of any caloric overload in postnatal life. However, in the total absence of INHBE/ Alk7 the effect on lipolysis becomes dominant with consequent build-up of ectopic fat and insulin resistance. While it is not entirely clear why subcutaneous, rather than visceral adipocytes would be favoured by PPARγ activity, it is worthy of note that the administration of drugs activating PPARγ preferentially promotes subcutaneous fat deposition[61].

Heterozygous LoF in *PPARG* has previously been linked to familial partial lipodystrophy, characterised by the loss of

subcutaneous fat from the extremities and metabolic abnormalities such as insulin resistance and hypertriglyceridemia[24–27]. However, *PPARG* has also been linked to placental development and angiogenesis[30]. Here, we saw a poor correlation between the variants' lipodystrophy-causing potential and effect on birth weight, suggesting the birth weight effect could be driven by another mechanism. Given the role of *PPARG* in placental development, it is possible that *PPARG* LoF variants act via the placenta to exert an effect on birth weight. Our results are consistent with recent observations of a high rate of small-for-gestational-age birth weights in patients who had familial partial lipodystrophy due to inherited *PPARG* mutations and who were not exposed to maternal diabetes in utero[28]. *PPARG* mutation carriers were also more likely to be born preterm, regardless of maternal diabetes status, which further supports the role of fetal *PPARG* in placental development. Data on gestational duration was not available for the UK Biobank participants, and it is possible that shorter gestation in rare *PPARG* variant carriers contributed to the associations we observed with birth weight. We note that the common fetal variant rs1801282 in *PPARG*, which is associated with type 2 diabetes risk in adulthood[62] via a fat distribution-mediated form of insulin resistance[63], is weakly associated with lower birth weight ($p = 0.0259$)[5] and was not associated with gestational duration in a fetal GWAS of gestational duration in 84,689 individuals ($p = 0.487$)[64]. Our observed associations between birthweight and *ADAMTS8*, *NRK*, *NYNRIN*, and *NOS3* are also likely driven by placental mechanisms, given the known roles of these genes[36–44,46].

Our study has several strengths, including a large discovery sample, an independent replication sample, support from common variant associations proximal to four of the genes, and approximate adjustment for correlation between maternal and fetal genotype. However, there are limitations. The sample size available for replication was limited, and although we observed strong evidence for consistent effect estimates, we recommend confirmation in larger datasets as they become available, along with proper conditional analyses in sufficiently-powered samples of mother-child pairs. The approximate adjustment for the correlation between maternal and fetal genotype used here is based on the strong assumption that all variants with a given annotation have equal effects on birth weight[5], which may not be the case. We also acknowledge the limited diversity of our samples. Studies of rare variants in well-powered samples of diverse ancestries are a priority for future research. While our study was underpowered to detect very rare pathogenic variants that have known birth weight effects[2], our sensitivity analysis of *GCK* showed that the known maternal and fetal associations were detectable by our method, despite the inevitable lower sensitivity and specificity of the exome-wide approach, and despite the lack of adjustment for opposing effects of maternal and fetal genotypes.

Overall, this study advances our understanding of the fetal and maternal genetic underpinnings of birth weight, providing strong evidence for causal genes and insights into biological pathways which are important targets for future research aiming to understand fetal growth and its links with long-term health.

## Methods

This research has been conducted using the UK Biobank Resource under Application Numbers 9905 and 7036. The UK Biobank study was undertaken with ethical approval from the North West Multicentre Research Ethics Committee (MREC) as a Research Tissue Bank (RTB).

Use of the Icelandic data was approved by the National Bioethics Committee (VSN-15-169). All genotyped participants signed a written informed consent allowing the use of their samples and data in projects at deCODE genetics approved by the NBC. Data was anonymized and encrypted by a third-party system, approved and monitored by the Icelandic Data Protection Authority[65].

### Exome wide association analyses in the UK Biobank

Analyses of the UK Biobank data were performed on the UK Biobank Research Analysis Platform (RAP; https://ukbiobank.dnanexus.com/). Birth weight was derived in the 453,505 UK Biobank participants with genetically defined European ancestry, as follows. Own birth weight (fetal genotype analysis) was derived using self-reported data (field 20022), while birth weight of the first child in UK Biobank women (maternal genotype analysis) was taken from hospital records and self-reports (fields 41284 and 2744). Data from each field were converted to kg (if in another unit) and if both were available, hospital records were preferentially retained for maximum accuracy. For both phenotypes, repeated responses across the assessment centre visits were combined by calculating average values. Individuals from multiple births (field 1777), who reported substantially different values between visits (>1 kg difference), or extreme birth weight values (<1 or ≥7 kg) were excluded from downstream analyses. After these exclusions, fetal birth weight data was available for 252,329 individuals (234,675 with exome and covariate data) and maternal birth weight data was available for the first child of 195,653 women (181,883 with exome and covariate data).

We conducted gene burden tests for both fetal and maternal birth weight, by combining effects of all rare (MAF < 0.1%) variants with predicted deleterious functional consequences across all protein-coding genes, as described in detail in Gardner et al.[56]. Briefly, we queried population-level VCF files with data for 454,787 individuals provided from the UK Biobank study. Using bcftools[66] multi-allelic variants were split and left-normalised, and all variants filtered using a missingness based approach. SNV genotypes with depth <7 and genotype quality < 20 or InDel genotypes with a depth < 10 and genotype quality < 20 were set to missing. We also tested for an expected reference and alternate allele balance of 50% for heterozygous SNVs using a binomial test; SNV genotypes with a binomial test $p.$ value $\leq 1 \times 10^{-3}$ were set to missing. Following genotype filtering, variants with > 50% missing genotypes were excluded from further analysis. Variants were then annotated with the ENSEMBL Variant Effect Predictor (VEP[67], v104) with the 'everything' flag and the LOFTEE plugin[68]. For each variant we prioritised a single MANE (v0.97) or VEP canonical ENSEMBL transcript and most damaging consequence as defined by VEP defaults. For the purposes of defining Protein Truncating Variants (PTVs), we grouped stop gained, splice donor/acceptor, and frameshift consequences. To define 'high-confidence' PTVs, we used the LOFTEE algorithm[68]. All variants were subsequently annotated using CADD (v1.6[69],). We then assessed the association between rare variant burden and birth weight, using BOLT-LMM (v2.3.5[70],) and a set of dummy genotypes representing per-gene carrier status, under two overlapping variant collapsing masks; i) high-confidence protein truncating variants (denoted PTV) and ii) High-confidence PTVs plus missense variants with CADD scores ≥ 25 (denoted Missense + PTV). All analyses were controlled for age, age$^2$, the first ten genetic ancestry principal components as calculated in Bycroft et al.[71], WES batch, and sex when running sex-combined analyses. We excluded genes with < 10 carriers of variants per mask.

This led to testing 16,735 genes with qualifying Missense + PTV variants and 13,684 with PTVs for fetal birth weight, and 17,745 with Missense + PTV variants and 13,968 with PTVs for maternal birth weight. The exome-wide significance thresholds were thus set to $p = 1.64 \times 10^{-6}$ (0.05/(16735 + 13684)) for fetal birth weight and $p = 1.58 \times 10^{-6}$ (0.05/(17745 + 13968)) for maternal birth weight.

### Confirmatory gene burden analyses

To replicate the findings from the above analysis and account for potential bias arising from only using one discovery approach, a second team independently derived the birth weight phenotypes for the maternal and fetal genetic analyses and conducted additional burden association analyses, specifically for the significantly associated birth weight genes.

Own birth weight (fetal genotype analysis) was derived using self-reported data (field 20022), excluding individuals who were part of a multiple birth (field 1777). Offspring birth weight (maternal genotype analysis) was derived from self-reported weight of first child (field 2744). Where individuals reported their birth weight at more than one assessment centre visit, reported birth weights were averaged unless the weights differed by ≥1 kg, in which case they were excluded. Own and offspring birth weight were available in 269,921 (232,876 with exome and covariate data) and 216,798 (152,585 with exome and covariate data) individuals respectively.

Multiallelic variants were similarly split and left-normalised. Variants flagged for exclusion by UK Biobank were removed, and remaining variants were annotated with their functional consequence in canonical transcripts using the Ensembl Variant Effect Predictor and the CADD and LOFTEE plugins (https://github.com/konradjk/loftee). Association testing was performed using REGENIE (v3.1.3[72],), and variants were grouped according to two collapsing masks; i) loss of function variants defined by LOFTEE as high confidence (denoted PTV) and ii) LoF plus missense SNVs or inframe insertions or deletions with CADD scores ≥ 25 (denoted Missense + PTV). All analyses were controlled for maternal age at first birth (field 3872) or year of birth, sex (for fetal birth weight), assessment centre, WES sequencing batch, and the first ten principal components.

Exome-wide significantly associated genes from the above BOLT-LMM analysis were queried in the REGENIE results.

### Follow-up analyses of identified genes

To distinguish between fetal genotyping-acting ('fetal-acting') and maternal genotype-acting ('maternal-acting') effects at each birth weight-associated gene, we applied a weighted linear model (WLM) to the burden test results. WLM estimates associations of fetal genotype conditional on maternal genotype, and vice versa, without the need for data from mother-child pairs. The WLM approach has been described previously[5].

To investigate potential sexual dimorphism, we performed sex-stratified burden tests using generalised linear models in the discovery sample and for the most strongly associated mask-gene combination. Sex-stratified effect estimates for each gene were compared using two-sample T-tests. Associations were deemed dimorphic if the p-value for the T-test was < 0.05.

Where described, further missense variant collapsing masks and leave-one out analyses were tested using BOLT-LMM or linear models in the same individuals and birth weight outcomes as the discovery analyses. Such masks were defined using a combination of CADD[69] and REVEL[73].

Conditional analyses where the effect of the identified rare exome variant collapsing masks was conditioned on the genotype at common birth weight GWAS signals were also tested using linear models in the same individuals and birth weight outcomes as the discovery analyses. Genotypes at GWAS signals (as identified in ref. [6]) were extracted in this discovery sample using plink (v1.90b6.18[74],). Birthweight outcomes were regressed against the covariates used in discovery analyses and the concurrent effects of exome variant carriage and GWAS signal genotype were tested against the residual birth weight variance. A change in effect estimate of more than 10% was considered to show significant attenuation in the presence of the GWAS variants.

### Related trait PheWAS

For all identified exome associations with birth weight, we performed small-scale PheWAS on a few predefined relevant phenotypes. We selected 10 phenotypes known to be associated with birth weight in epidemiological studies and for which we had a well-powered sample in UK Biobank: measures of body size and adiposity in adulthood (BMI, Body fat %, WHR adjusted for BMI, Height); measures of growth and body size in childhood (Comparative height at age 10, Comparative

size at age 10, IGF-1); blood pressure (systolic and diastolic), and type 2 diabetes.

For BOLT-LMM analyses, phenotypic outcomes were defined as follows. BMI raw values were used from field 21001. Body fat % was extracted using data from field 23099. Data from subsequent visits were used, if missing for a given instance. Comparative height at age 10 was recoded to have individuals who in field 1697 reported being 'Shorter' as 0, 'About average' as 1, and 'Taller' as 2. Individuals who reported 'Do not know' or 'Prefer not to answer' were set to 'NA'. Following variable recoding, this phenotype was run as a continuous trait. Comparative size at age 10 was defined as above using data from field 1687. Systolic and diastolic blood pressure was derived using data from fields 4080 and 4079, accordingly. For each, the average blood pressure at each assessment visit was calculated over the two available measurements and data from subsequent visits were used, if missing for a given instance. Individuals indicating the use of blood pressure medications in fields 6153 and 6177 were adjusted for in the association models. Height raw values were used from field 50. IGF-1 levels were derived using data from field 30770, with exclusion of individuals > 5 standard deviations from the mean. T2D was derived as described in[56] and using data from fields 4041, 10844, 2443, 6177, 6153, 20002, 20003, 41202, 41204, 40001and 40002. WHR adjusted for BMI; WHR was calculated using data from field 48 and 49. > 4 standard deviations from the mean excluded following initial calculation and prior to adjusting with BMI (field 21001). Paired BMI and WHR data from subsequent visits were used, if missing for a given instance.

For REGENIE analyses, phenotypic outcomes are as follows: BMI was taken from the baseline value for field 21001 and residualised on age at assessment (field 21003), genetic sex (field 22001), assessment centre (field 54) and PCs 1-5. Blood pressure measurements were taken from the automated readings for diastolic blood pressure (field 4079) and systolic blood pressure (field 4080). Where two blood pressure readings were available, the average between the two was taken, excluding individuals where the measurements differed by > 4.56 SD. Individuals with a blood pressure > .56 SD away from the mean were excluded. Blood pressure medication was accounted for by adding 10 and 15 mmHg to diastolic and systolic blood pressure, respectively. Height raw values were taken from baseline values for field 50, where people whose height was 100 cm and sitting height (field 20015) was 20 cm, were removed. Raw IGF-1 values were used from field 30770. Size at age 10 was based on a continuous simulation of childhood BMI. The simulation converted the categorical variables 'thinner', 'the same size' or 'plumper' in field 1687 to a continuous distribution based on the number of individuals selecting each category. The full methods for the simulation have previously been reported[75]. WHR was calculated using waist circumference (field 48) and hip circumference (field 49). If a follow-up visit was recorded, the WHR was taken from the follow-up data. Otherwise, it was taken from the first recorded WHR measurement.

The multiple testing threshold was set at p < 0.005, after correcting for the above 10 queried phenotypes.

### GWAS associations at identified loci

GWAS signals for fetal and maternal birth weight were accessed from Juliusdottir et al.[6]. These signals were lifted back to GRCh37 annotated with all proximal genes, defined as those within 300 kb up- or downstream of the genes start or end sites, using the NCBI RefSeq gene map for GRCh37 (via http://hgdownload.soe.ucsc.edu/goldenPath/hg19/database/). For signals proximal to the exome-associated genes we assessed whether the exome-associated gene was the closest gene to the GWAS signal. These signals were also queried using eQTL data from the GTEx study (V8[10],) and plotted using LocusZoom[76].

We performed gene-centric biological pathway enrichment analysis using g:Profiler (via the R client gprofiler2[77], v0.2.1, accessed on 23/11/2023). Pathway enrichment analyses were performed using the

genes calculated to be the closest per signal to the GWAS signals from Juliusdottir et al.[6]. Pathways with a Bonferroni corrected *p*-value < 0.05 were considered significantly enriched in the birth weight GWAS.

We also examined all genes within 300 kb of the previously published birth weight loci from Juliusdottir et al. For this restricted set of loci previously shown to have common genetic associations with birth weight, we applied a Benjamini-Hochberg procedure to identify genes with evidence of association.

### Replication of exome associations in deCODE

**Data preparation.** The genome of the Icelandic population was characterised by whole-genome sequencing (WGS) of 63,460 Icelanders using Illumina standard TruSeq methodology to a mean depth of 35x (SD 8x) with subsequent long-range phasing[78], and imputing the information into 173,025 individuals chip-genotyped employing multiple Illumina platforms[79]. Variant calling was performed using GraphTyper[80] (v2.6) and chip data was phased using SHAPEIT4[81].

We used Variant Effect Predictor (VEP[67]) to attribute predicted consequences to the variants sequenced in each dataset. We classified as high-impact variants those predicted as start-lost, stop-gain, stop-lost, splice donor, splice acceptor or frameshift, collectively called loss-of-function (LoF) variants. Variants of moderate impact are those classified as missense, splice-region and in-frame-indels.

Information on birthweight comes from the Icelandic birth register and Primary Health Care Clinics of the Capital area. Individual's fetal birth weight was available in up to 36,578 genotyped and 11,184 WGS participants, while maternal birth weight was available in 45,622 genotyped and 17,177 WGS participants.

**Gene burden associations.** We defined different models to group together various types of variant: (i) only LoF variants, filtered with LOFTEE[68]; (ii) LoF and MIS (as described on Variant annotation), when predicted deleterious by CADD[69] score ≥ 25. In all models, we used MAF < 0.1% to select variants for analyses.

Quantitative traits were analysed using a linear mixed model implemented in BOLT-LMM[70]. To estimate the quality of the sequence variants across the entire set we regressed the alternative allele counts (AD) on the depth (DP) conditioned on the genotypes (GT) reported by GraphTyper[82]. For a well behaving sequence variant, the mean alternative allele count for a homozygous reference genotype should be 0, for a heterozygous genotype it should be DP/2 and for homozygous alternative genotype it should be DP. Under the assumption of no sequencing or genotyping error, the expected value of AD should be DP conditioned on the genotype, in other words an identity line (slope 1 and intercept 0). Deviations from the identity line indicate that the sequence variant is spurious or somatic. We filter variants with slope less than 0.5. Additionally, GraphTyper employs a logistic regression model that assigns each variant a score (AAscore) predicting the probability that it is a true positive. We used only variants that have a AAscore > 0.8.

### Analysis of birth weight associations with pathogenic *GCK* mutations

We used a similar procedure to that used to define pathogenic variants in GCK in a recent paper[83]. We reviewed all heterozygous missense/PTV variants in UK Biobank participants with whole exome sequencing that were observed at minor allele frequency (MAF) < 0.001. We included variants in the analysis if missense/PTV variants were classified as pathogenic or likely pathogenic based on ACMG/AMP guidelines by clinical scientists at Exeter Molecular Genetic laboratory as part of routine clinical diagnostic care (i.e., previously seen in the MODY probands)[35] and were ultra-rare in the population (maximum allele count of 2 in gnomADv2.1, MAF < $1.4 \times ^{-5}$). We manually reviewed sequence read data for all the pathogenic variants (missense and PTVs) in Integrative Genomics Viewer (IGV) to remove false-positive variants. We used linear regression to assess the difference in birth weight

associated with being a carrier (separate maternal and fetal analyses) and adjusted for genotype, sex, mother's diabetes status, batch, year of birth, centre and principal components in the fetal analysis and genotype, batch, age at first birth, centre and principal components in the maternal birth weight analysis. We then compared the estimated effects and 95% confidence intervals for the pathogenic variants with those obtained from our PTV and Missense + PTV masks in the REGENIE analyses.

### Reporting summary

Further information on research design is available in the Nature Portfolio Reporting Summary linked to this article.

## Data availability

This research has been conducted using the UK Biobank Resource under application numbers 9905 and 7036. Data from the UK Biobank are available by application to all bona fide researchers in the public interest at https://www.ukbiobank.ac.uk/enable-your-research/apply-for-access. Additional information about registration for access to the data is available at www.ukbiobank.ac.uk/register-apply/. Sequence variants from the Icelandic population whole-genome sequence data have been deposited at the European Variation Archive under accession PRJEB15197. In accordance with the Icelandic National Bioethics Committee approval, individual level genotype or phenotype data cannot be accessed. Birth weight GWAS data from[6] used in this study are available at https://www.decode.com/summarydata/. Any remaining data are contained within the article and/or its Supplementary Information Files.

## Code availability

Code for WES data processing and association testing is available on GitHub (https://github.com/mrcepid-rap/mrcepid-runassociationtesting). No custom code was developed for this study.

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

## Acknowledgements

K.A.K., L.R.K., E.J.G., Y.Z., F.R.D., K.K.O. and J.R.B.P. are supported by the Medical Research Council (MC_UU_00006/2). B.E.M.L., R.N.B. and R.M.F. were supported by a Wellcome Senior Research Fellowship (WT220390). This research was funded in part by the Wellcome Trust (WT220390). R.M.F. is also supported by a grant from the Eunice Kennedy Shriver National Institute Of Child Health & Human Devel-opment of the National Institutes of Health under Award Number R01HD101669. This study was supported by the National Institute for Health and Care Research (NIHR) Exeter Biomedical Research Centre and the NIHR Cambridge Comprehensive Biomedical Research Centre. The views expressed are those of the authors and not necessarily those of the NIHR or the Department of Health and Social Care. For the purpose of open access, the authors have applied a CC BY public copyright licence to any Author Accepted Manuscript version arising from this submission.

## Author contributions

K.A.K., B.E.M.L., L.R.K., K.K.O., R.N.B., J.R.B.P., and R.M.F. analysed and interpreted the original data and wrote the manuscript. V.S., V.T., T.O., G.T., and K.S. contributed the Icelandic replication data. L.S., K.A.P., G.H., E.J.G., A.R.W., Y.Z., F.R.D., S.E.O., A.T.H., and S.O. contributed to data analysis and/or interpretation. All authors revised, critically reviewed and approved the manuscript.

## Competing interests

J.R.B.P. and E.J.G. are employees/shareholders of Insmed. J.R.B.P. also receives research funding from GSK and consultancy fees from WW International. Y.Z. is a UK University worker at GSK. S.O. has undertaken remunerated consultancy work for Pfizer, Third Rock Ventures, AstraZeneca, NorthSea Therapeutics and Courage Ther-apeutics. V.S., V.T., T.O., G.T., and K.S. are employees of deCODE genetics, a subsidiary of Amgen. The remaining authors declare no competing interests.

## Additional information

Katherine A. Kentistou [1,7], Brandon E. M. Lim[2,7], Lena R. Kaisinger[1,7], Valgerdur Steinthorsdottir [3], Luke N. Sharp [2],
Kashyap A. Patel [2], Vinicius Tragante [3], Gareth Hawkes [2], Eugene J. Gardner [1], Thorhildur Olafsdottir [3],
Andrew R. Wood [2], Yajie Zhao [1], Gudmar Thorleifsson [3], Felix R. Day [1], Susan E. Ozanne[4], Andrew T. Hattersley [2],
Stephen O'Rahilly [4], Kari Stefansson [3,5], Ken K. Ong[1,6,7], Robin N. Beaumont [2,7], John R. B. Perry [1,4,7] &
Rachel M. Freathy [2,7] ✉

¹MRC Epidemiology Unit, Box 285 Institute of Metabolic Science, University of Cambridge School of Clinical Medicine, Cambridge, UK. ²Department of
Clinical and Biomedical Sciences, Faculty of Health and Life Sciences, University of Exeter, Exeter, UK. ³deCODE genetics/Amgen, Inc., 102 Reykjavik,
Reykjavik, Iceland. ⁴MRC Metabolic Diseases Unit, Institute of Metabolic Science, University of Cambridge School of Clinical Medicine, University of Cam-
bridge, Cambridge, UK. ⁵Faculty of Medicine, University of Iceland, Reykjavik, Iceland. ⁶Department of Paediatrics, University of Cambridge, Cambridge, UK.
⁷These authors contributed equally: Katherine A. Kentistou, Brandon E. M. Lim, Lena R. Kaisinger, Ken K. Ong, Robin N. Beaumont, John R. B. Perry, Rachel M.
Freathy. ✉e-mail: R.Freathy@exeter.ac.uk

