## [Transparent Peer Review file · Nature Communications]

Rare variant associations with birth weight identify genes involved in adipose tissue regulation, placental function and insulin-like growth factor signalling

Corresponding Author: Professor Rachel Freathy

Version 0:

Reviewer comments:

Reviewer #1

(Remarks to the Author)

This is a nice paper with a clean and simple design that produces some new findings. I have mostly minor comments:

1. It seems like many of the genes discussed are already previously identifiable from GWAS. It would be good to clarify in the discussion and gene description sections which insights are newly enabled by these findings versus which ones were already known versus which ones were highly suspected but not confirmed. For example, how much of the IGF1R insight on lines 343-355 could not have been made from previous GWAS results alone? Similar things could be said about the other gene-specific paragraphs.

2. Why was 500kb used to determine if a gene was near a GWAS signal but 300kb used to define genes for the FDR analysis?

3. In the gene set enrichment paragraph (particularly, when the exome enrichments are compared to the GWAS enrichments), how much of the enrichment was driven by the top gene associations -- in particular, how much by the top gene associations in GWAS genes? It would seem that the GWAS and exome results are not fully independent, and therefore it is unclear how much *additional* support the gene set analysis adds to the validity of the associations beyond the already described overlap with GWAS loci. This is even more pertinent if the FDR analysis genes are going into the enrichment analysis, since those were ascertained on proximity to GWAS loci -- did those go into the analysis, and if so how was this ascertainment bias adjusted for?

4. What was the rationale for the 10 traits chosen for the pheWAS of secondary traits?

5. I didn't follow the rationale for the GCK analysis

6. The PPARG missense variant analysis is a little hard to understand because it seems like the lack of association with adipocyte affecting variants is used to suggest a potential placenta-mediated effect. Is this what is being suggested?

Reviewer #2

(Remarks to the Author)

There continues to be much interest in assessing fetal and maternal genetic, epigenetic and non-genetic contributions to fetal growth as abnormal birth weight (low or high) for gestational age are often associated with cardio-metabolic diseases in adult years. The availability of large and smaller databases (EGG Consortium, UKBB; Icelandic birth registry, etc) with birth weight information (albeit mostly self-reporting for EGG and UKBB), and new algorithms (e.g. WLM), had led to multiple GWAS reports within the last decade that have revealed >200 regions of the genome where common SNPs are associated with birth weight.

In this well-written report, the authors identify 8 genes likely to be causally related to birth weight by performing ExWAS for

rare variants of gene burden using the UKBB database, observed directionally concordant associations in the Icelandic database, and demonstrated that these genes fell within the contributing regions identified by GWAS. Two of the 8 also showed maternal effects. An additional gene (ADAMTS8) showed only maternal effects. This surprisingly small number of genes is due, in part, to stringent filtering criteria, including selecting rare exonic SNPs that resulted in protein truncation (PTV) or predicted damaging missense variants. Both PTV and selected missense variants would be expected to disrupt protein expression, and, by extension, disrupt biological processes. It is of note that analysis of these databases inevitably are biased towards genes that act in an autosomal dominant manner, either by inducing haploinsufficiency or hypomorphic states.

The 8 identified genes are involved in biological processes that, perhaps unsurprisingly, could well modulate birth weight: growth and metabolism (IGF1R, PAPP2), adipose tissue regulation (PPARG, INHBE, ACVR1C), fetoplacental angiogenesis/hypertension (PPARG, NOS3, NRK, NYNRIN). The inclusion of biological implications analyses such as pathway enrichment analyses and gene burden associations with related traits, supported how each may contribute towards birth weights. To test the sensitivity of their ExWAS approach, they successfully applied the analysis on well documented pathogenic variants within the GCK genes known to affect birth weight.

(1) Overall, the application of ExWAS approach with phenotypic associations in birthweight-related traits was thorough and logical. The genes revealed in this study provides confirmation that these genes are likely to be involved in birth weight regulation, either based on previous association studies from the largest and most recent birth weight GWAS meta-analysis, or based on previous clinical phenotypic reports (e.g. IGF1R, PAPP2). However, no new genes were revealed that can influence birthweight which reduced the novelty of this study.

(2) The self-reporting of birth weight and lack of gestational age information, are bothersome as individuals may be biased in reporting birth weight, and gestational age can also influence interpretation of whether the birth weight is considered clinically normal. For example, birth weight at gestational age (GA) greater than 37 weeks may be low, but, using, for example, Fenton Scoring method for neonates with GA <37 weeks, the birth weight may be normal. Perhaps the discovery size of the UKBB may negate these concerns.

(3) The strengths and weaknesses of this study were nicely discussed.

(4) The ExWAS analysis integrated with PheWAS: inevitably favor autosomal dominant variants, which may well miss relevant variants in genes that are autosomal recessive and have biological impacts in homozygous or compound heterozygous or even mosaic states.

(5) There is a 2022 Mol Biol Evol paper that suggest NYNRIN may be involved in placental emergence: PMID: 35959649 PMID: PMC9447858 DOI: 10.1093/molbev/msac176

Version 1:

Reviewer comments:

Reviewer #1

(Remarks to the Author)

I thank the authors for their response to my comments.

I think that their response:

"However, we disagree that any of the genes highlighted here were 'previously known' from GWAS. Linkage disequilibrium at common GWAS loci means that identifying the causal gene or variant is only an (often uncertain) inference, even when a GWAS locus overlies a plausible candidate gene (see for example studies aiming to understand the causal gene at the FTO locus; 10.1056/NEJMoa1502214)"

misses the forest for the trees. True, it is possible that these genes were not the GWAS effector genes despite being the nearest genes, and it is true that sometimes full functional workup is needed to identify the causal gene. However, in this case my comment was to ask them to clarify what new biology was learned from the present study that was not apparent from the GWAS. All four of these genes have either mouse data or pathway data that made them very strong candidates at these loci.

The exome data is valuable confirmation of these genes as causal, and validates the hypothesis that careful GWAS analysts would have made after identifying SNPs near these genes. I am just asking for the authors to clearly state this as the major advance.

However, my original comment was not intended to be negative and also sought to motivate potentially more analyses that can be done with exome data -- for example, does an allelic series of rare variants in the gene, or the direction of effect of damaging variants in the gene, offer additional insights beyond gene identification.

Reviewer #2

(Remarks to the Author)

My concerns are addressed for the most part.

However, regarding IGF1R, the authors should be aware that many IGF1R variants have been clinically reported, and modify Discussion accordingly:

The two references quoted in this study for IGF1R (10.3889/oamjms.2018.416; 10.1056/NEJMoa010107) specifically used SGA (small for gestational age, birth weight and/or birth length below normal for gestational age) as one selective criteria for cohorts studied, for which it is not surprising that only a couple of IGF1R defects were identified. To date, more than 60 IGF1R exonic variants (majority in heterozygous state) have been clinically reported (reviewed <https://doi.org/10.1007/s11154-020-09603-3>, section 4.1) most of which are associated with SGA. It is, therefore, not a surprise that IGF1R was one of the genes identified in this report – i.e. this report supports clinical findings.

I do find it surprising that the IGF1 and IGF2 genes which, when defective, are associated with low birth weight, were not amongst the identified genes – perhaps too few variants in database analyzed.

Responses to reviewers' comments

Response: We are grateful to the reviewers for their time and helpful comments. Reviewers' comments are in blue below with point-by-point responses following each comment. All new changes to the manuscript have been marked in red.

Reviewer #1

This is a nice paper with a clean and simple design that produces some new findings. I have mostly minor comments:

1. It seems like many of the genes discussed are already previously identifiable from GWAS. It would be good to clarify in the discussion and gene description sections which insights are newly enabled by these findings versus which ones were already known versus which ones were highly suspected but not confirmed. For example, how much of the *IGF1R* insight on lines 343-355 could not have been made from previous GWAS results alone? Similar things could be said about the other gene-specific paragraphs.

Response: We thank the reviewer for their positive comments. However, we disagree that any of the genes highlighted here were 'previously known' from GWAS. Linkage disequilibrium at common GWAS loci means that identifying the causal gene or variant is only an (often uncertain) inference, even when a GWAS locus overlies a plausible candidate gene (see for example studies aiming to understand the causal gene at the *FTO* locus; 10.1056/NEJMoa1502214).

Of our 9 identified genes, 4 (*IGF1R*, *PAPPA2*, *ACVR1C*, and *NOS3*) are proximal to lead GWAS SNPs for birth weight, as described in Juliusdottir et al. However, all but one of these lead SNPs were intronic, limiting causal gene identification at these GWAS loci. A key contribution of our study, therefore, is to provide strong and independent evidence that variation in these 4 genes causally impacts birth weight. Linking rare, deleterious variants to variation in birth weight removes uncertainty around gene causality and is novel and important data. None of the specific biological inferences presented here were highlighted in previous GWAS efforts, due to the above limitations on identification of causal genes from GWAS.

In reply to the reviewer's specific query on *IGF1R*, our findings were not informed by previous GWAS, but by our recent Type 2 diabetes rare variant analysis (Gardner et al. 2022). Together, these show that rare fetal LOF variants in *IGF1R* confer lower birth weight and also higher Type 2 diabetes risk, which provide new insights into the mechanisms that underpin the widely reported phenotypic associations between early growth and later metabolic disease risks. See also our per gene summary in response to Reviewer 2's first comment below.

We have edited the discussion to re-iterate which genes had been suggested by GWAS loci (lines 329-330), and also to highlight the novel insights on *IGF1R* (lines 351-355).

2. Why was 500kb used to determine if a gene was near a GWAS signal but 300kb used to define genes for the FDR analysis?

Response: We agree that this difference was unsubstantiated and potentially confusing. We have now updated all analyses using 300kb windows.

3. In the gene set enrichment paragraph (particularly, when the exome enrichments are compared to the GWAS enrichments), how much of the enrichment was driven by the top gene associations -- in particular, how much by the top gene associations in GWAS genes? It would seem that the GWAS and exome results are not fully independent, and therefore it is unclear how much *additional* support the gene set analysis adds to the validity of the associations beyond the already described overlap with GWAS loci. This is even more pertinent if the FDR analysis genes are going into the enrichment analysis, since those were ascertained on proximity to GWAS loci -- did those go into the analysis, and if so how was this ascertainment bias adjusted for?

Response: We used the pathway enrichment analysis to test whether birth weight genes implicated by GWAS loci were enriched in similar pathways highlighted by our exome-identified birth weight genes (insulin and growth factor response and angiogenesis), and found supportive evidence of such enrichment.

We recognise that some genes are shared across the exome and GWAS data and we now add a sentence to clarify this (lines 174-175). However, these lines of evidence are fully independent as we show that our exome gene-burden associations are independent of the GWAS signals at the corresponding loci (lines 155-159). Even if we remove the exome-identified genes from the list of GWAS genes entered into the pathway analysis, we still see enrichment for genes in the "insulin secretion" (before $p=0.0232$ and after exclusion $p=0.0202$) and "circulatory system development" pathways (before $p=0.00289$ and after exclusion $p=0.0363$). However, such an approach is needlessly stringent.

Genes from the FDR analysis were not used to support the above pathway analysis findings.

4. What was the rationale for the 10 traits chosen for the pheWAS of secondary traits?

Response: The 10 traits tested in the PheWAS analysis represent traits with widely reported birth weight associations in large phenotypic studies. These included measures of: adult body size/adiposity (BMI, Body fat %, WHR adjusted for BMI, Height), child growth/body size (Comparative height at age 10, Comparative size at age 10, IGF-1), and cardiometabolic disease risk (systolic and diastolic blood pressure and Type 2 diabetes). We have added an explanation of this to the methods (lines 594-599).

5. I didn't follow the rationale for the GCK analysis

Response: We selected *GCK* for a sensitivity analysis because it is known to harbour pathogenic variants that influence birth weight when carried by either mother or child. Our aim was to assess how bioinformatically-informed variant classification, as used in our and others'

exome-wide association studies, compares with clinical annotation of known pathogenic variants. Rare pathogenic variants in *GCK* are a known cause of monogenic diabetes: carriers have mild, stable fasting hyperglycemia throughout life. Although rare (1.1 in 1000 individuals [Chakera et al 2014, Diabetes Care]), pathogenic variants in *GCK* are (to our knowledge) the most common, clinically-annotated variants with established impacts on birth weight when carried by the mother (via effects on maternal fasting glucose) or the fetus/child (via effects on fetal insulin) and, with sufficient carriers in UK Biobank to detect statistically significant associations with birth weight. We have added this rationale on lines 301-306.

6. The *PPARG* missense variant analysis is a little hard to understand because it seems like the lack of association with adipocyte affecting variants is used to suggest a potential placenta-mediated effect. Is this what is being suggested?

Response: We have clarified the text to explain that our aim was to examine if the BW effects of *PPARG* might correspond to its known effects on adipogenesis (line 259-260). Our lack of such evidence indicates that data is needed on alternative mechanisms, such as placental morphology and function (clarified on lines 273-275).

Reviewer #2

There continues to be much interest in assessing fetal and maternal genetic, epigenetic and non-genetic contributions to fetal growth as abnormal birth weight (low or high) for gestational age are often associated with cardio-metabolic diseases in adult years. The availability of large and smaller databases (EGG Consortium, UKBB; Icelandic birth registry, etc) with birth weight information (albeit mostly self-reporting for EGG and UKBB), and new algorithms (e.g. WLM), had led to multiple GWAS reports within the last decade that have revealed >200 regions of the genome where common SNPs are associated with birth weight.

In this well-written report, the authors identify 8 genes likely to be causally related to birth weight by performing ExWAS for rare variants of gene burden using the UKBB database, observed directionally concordant associations in the Icelandic database, and demonstrated that these genes fell within the contributing regions identified by GWAS. Two of the 8 also showed maternal effects. An additional gene (*ADAMTS8*) showed only maternal effects. This surprisingly small number of genes is due, in part, to stringent filtering criteria, including selecting rare exonic SNPs that resulted in protein truncation (PTV) or predicted damaging missense variants. Both PTV and selected missense variants would be expected to disrupt protein expression, and, by extension, disrupt biological processes. It is of note that analysis of these databases inevitably are biased towards genes that act in an autosomal dominant manner, either by inducing haploinsufficiency or hypomorphic states.

The 8 identified genes are involved in biological processes that, perhaps unsurprisingly, could well modulate birth weight: growth and metabolism (*IGF1R*, *PAPPA2*), adipose tissue regulation (*PPARG*, *INHBE*, *ACVR1C*), fetoplacental angiogenesis/hypertension (*PPARG*, *NOS3*, *NRK*, *NYNRIN*). The inclusion of biological implications analyses such as pathway enrichment analyses and gene burden associations with related traits, supported how each may contribute

towards birth weights. To test the sensitivity of their ExWAS approach, they successfully applied the analysis on well documented pathogenic variants within the GCK genes known to affect birth weight.

(1) Overall, the application of ExWAS approach with phenotypic associations in birthweight-related traits was thorough and logical. The genes revealed in this study provides confirmation that these genes are likely to be involved in birth weight regulation, either based on previous association studies from the largest and most recent birth weight GWAS meta-analysis, or based on previous clinical phenotypic reports (e.g. IGF1R, PAPP2). However, no new genes were revealed that can influence birth weight which reduced the novelty of this study.

Response: We thank the reviewer for their positive comments. However, we disagree that any of the genes highlighted here were 'previously known' from GWAS. Linkage disequilibrium at common GWAS loci means that identifying the causal gene or variant is only an (often uncertain) inference, even when a GWAS locus overlies a plausible candidate gene (see for example studies aiming to understand the causal gene at the FTO locus; 10.1056/NEJMoa1502214).

Of our 9 identified genes, 4 (*IGF1R*, *PAPP2*, *ACVR1C*, and *NOS3*) are proximal to lead GWAS SNPs for birth weight, as described in Juliusdottir et al. However, all but one of these lead SNPs were intronic, limiting causal gene identification at these GWAS loci. A key contribution of our study, therefore, is to provide strong and independent evidence that variation in these 4 genes causally impacts birth weight. Linking rare, deleterious variants to variation in birth weight removes uncertainty around gene causality and is novel and important data. None of the specific biological inferences presented here were highlighted in previous GWAS papers. Even where proximal genes were found, they were often not mentioned in the text of those GWAS papers, reflecting the above limitations on identification of causal genes from GWAS data.

See the following per gene summary:

ADAMTS8: No GWAS signal, no reported link to birth weight.

NYNRIN: No GWAS signal, no reported link to birth weight.

NRK: No GWAS signal, only 1 reported mouse study (cited) link to birth weight.

PPARG: No GWAS signal. No birth weight effect reported in most familial-partial lipodystrophy patients (e.g. 10.1007/s00125-021-05386-7) even though it is a topic of extensive clinical study. This limited clinical evidence is cited.

NOS3: Juliusdottir et al. identified a GWAS signal proximal to *NOS3* but did not further mention this gene. Conflicting reported associations in very small studies, e.g. 10.1007/s11033-020-05897-3.

PAPP2: Juliusdottir et al. identified a GWAS signal proximal to *PAPP2* but did not further mention this gene. Conflicting evidence from small clinical case reports are cited.

INHBE: No GWAS signal. LOF association with birth weight was reported by Deaton et al. (cited), which we build on with a splice variant leave-one-out analysis.

ACVR1C: Juliusdottir et al. identified a GWAS signal proximal to *ACVR1C*. LOF association with birth weight was also reported by Deaton et al., as above.

IGF1R: Juliusdottir et al. identified a GWAS signal proximal to *IGF1R* but did not further mention this gene. Some evidence from very small case reports, e.g. 2 cases in 10.3889/oamjms.2018.416, and 2 cases in 10.1056/NEJMoa010107 (summarised in the cited review - Ref 48).

In summary, most of the genes highlighted in this work represent novel and definitive links to normal variation in human birth weight.

(2) The self-reporting of birth weight and lack of gestational age information, are bothersome as individuals may be biased in reporting birth weight, and gestational age can also influence interpretation of whether the birth weight is considered clinically normal. For example, birth weight at gestational age (GA) greater than 37 weeks may be low, but, using, for example, Fenton Scoring method for neonates with GA <37 weeks, the birth weight may be normal. Perhaps the discovery size of the UKBB may negate these concerns.

Response: We tried to minimise error in self-reporting of birth weight by; 1) where available hospital recorded birth weight was prioritised over self-report, 2) we excluded individuals from analysis if their repeated self-reported birth weights (at different assessment visits) differed by >1kg, 3) calculated the average value if their repeated self-reported birth weights differed by <1kg, 4) excluded from analysis extreme birth weight values (<1 or ≥7kg).

We acknowledge that the lack of gestational age information is a limitation, and this is discussed on lines 417-419. We also show here a sensitivity analysis where (in addition to the above steps) we include only birth weight values between 2.5 to 4.5 kg (to avoid confounding by preterm delivery). Unsurprisingly, the associations are generally weaker due to the truncated distribution, but results for all genes are consistent with the primary analysis.

(3) The strengths and weaknesses of this study were nicely discussed.

Response: Thank you.

(4) The ExWAS analysis integrated with PheWAS: inevitably favor autosomal dominant variants, which may well miss relevant variants in genes that are autosomal recessive and have biological impacts in homozygous or compound heterozygous or even mosaic states.

Response: We agree that this population-based study design is unsuited to detect recessive effects due to the scarcity of homozygotes for rare variants. For example, of 932 carriers of DMG variants in *IGF1R* we found only 1 putative compound heterozygous individual, 1 out of 1971 *NOS3* variant carriers, and only 2 out of 1183 *PAPPA2* variant carriers.

(5) There is a 2022 *Mol Biol Evol* paper that suggest *NYNRIN* may be involved in placental emergence: PMID: 35959649 PMCID: PMC9447858 DOI: 10.1093/molbev/msac176

Response: Thank you. We now cite this paper in the Results and Discussion (lines 297-298 and 424).

Responses to reviewers' comments

Response: We are grateful to the reviewers for their time and helpful comments. Reviewers' comments are in blue below with point-by-point responses following each comment.

Reviewer #1

I thank the authors for their response to my comments.

I think that their response: "However, we disagree that any of the genes highlighted here were 'previously known' from GWAS. Linkage disequilibrium at common GWAS loci means that identifying the causal gene or variant is only an (often uncertain) inference, even when a GWAS locus overlies a plausible candidate gene (see for example studies aiming to understand the causal gene at the FTO locus; 10.1056/NEJMoa1502214)" misses the forest for the trees.

True, it is possible that these genes were not the GWAS effector genes despite being the nearest genes, and it is true that sometimes full functional workup is needed to identify the causal gene. However, in this case my comment was to ask them to clarify what new biology was learned from the present study that was not apparent from the GWAS. All four of these genes have either mouse data or pathway data that made them very strong candidates at these loci.

The exome data is valuable confirmation of these genes as causal, and validates the hypothesis that careful GWAS analysts would have made after identifying SNPs near these genes. I am just asking for the authors to clearly state this as the major advance. However, my original comment was not intended to be negative and also sought to motivate potentially more analyses that can be done with exome data -- for example, does an allelic series of rare variants in the gene, or the direction of effect of damaging variants in the gene, offer additional insights beyond gene identification.

Response: We thank the reviewer for the clarification and agree with their identification of one of the major advances of our paper, which we highlight on lines 278-281. We agree that an allelic series analysis could offer further insight, although it would lack power with the current dataset. We commented on insights arising from the direction of effect of the damaging variants we identified on lines 152-171, also in conjunction with other related phenotypes, and agree this offers additional insight.

Reviewer #2

My concerns are addressed for the most part.

However, regarding IGF1R, the authors should be aware that many IGF1R variants have been clinically reported, and modify Discussion accordingly: The two references quoted in this study for IGF1R (10.3889/oamjms.2018.416; 10.1056/NEJMoa010107) specifically used SGA (small for gestational age, birth weight and/or birth length below normal for gestational age) as one selective criteria for cohorts studied, for which it is not surprising that only a couple of IGF1R defects were identified. To date, more than 60 IGF1R exonic variants (majority in heterozygous state) have been clinically reported (reviewed <https://doi.org/10.1007/s11154-020-09603-3>, section 4.1) most of which are associated with SGA. It is, therefore, not a surprise that IGF1R was one of the genes identified in this report – i.e. this report supports clinical findings. I do find it surprising that the IGF1 and IGF2 genes which, when defective, are associated with low birth weight, were not amongst the identified genes – perhaps too few variants in database analyzed.

Response: We thank the reviewer for the additional information. We have added the above review and a further clinical case study discussing carriers with SGA to the section of the discussion which stated the evidence already known for *IGF1R* (lines 285-288). We note that the review describes several rare cases with homozygous mutations in *IGF1R* and low birthweight but the evidence in heterozygous carriers was previously sparse. We were not powered to detect altered birth weight in carriers of *IGF1* and *IGF2* variants in our analyses.